# Identification of Cancer Associated Fibroblasts Related Genes Signature to Facilitate Improved Prediction of Prognosis and Responses to Therapy in Patients with Pancreatic Cancer

**DOI:** 10.3390/ijms26104876

**Published:** 2025-05-19

**Authors:** Yong Zhou, Yanxi Lu, Franziska Czubayko, Jisheng Chen, Shuwen Zheng, Huaqing Mo, Rui Liu, Georg F. Weber, Robert Grützmann, Christian Pilarsky, Paul David

**Affiliations:** 1Department of Surgery, University Hospital Erlangen, Friedrich-Alexander Universität Erlangen-Nürnberg (FAU), 91054 Erlangen, Germany; yong.zhou@fau.de (Y.Z.); yanxi.lu@uk-erlangen.de (Y.L.); franziska.czubayko@uk-erlangen.de (F.C.); jisheng.chen@extern.uk-erlangen.de (J.C.); shuwen.zheng@extern.uk-erlangen.de (S.Z.); huaqing.mo@fau.de (H.M.); rui.liu@fau.de (R.L.); georg.weber@uk-erlangen.de (G.F.W.); robert.gruetzmann@uk-erlangen.de (R.G.); 2Deutsches Zentrum für Immuntherapie, Universitätsklinikum Erlangen, Friedrich-Alexander Universität Erlangen-Nürnberg (FAU), 91054 Erlangen, Germany; 3Bavarian Cancer Research Center (BZKF), 91052 Erlangen, Germany

**Keywords:** pancreatic cancer, cancer associated fibroblast, WGCNA, prognosis, chemotherapy, immunotherapy, molecular docking

## Abstract

Pancreatic cancer (PC) is highly aggressive, with a 5-year survival rate of 12.8%, making early detection vital. However, non-specific symptoms and precursor lesions complicate diagnosis. Existing tools for the early detection of PC are limited. CAFs are crucial in cancer progression, invasion, and metastasis, yet their role in PC is poorly understood. This study analyzes mRNA data from PC samples to identify CAF-related genes and drugs for PC treatment using algorithms like EPIC, xCell, MCP-counter, and TIDE to quantify CAF infiltration. Weighted gene co-expression network analysis (WGCNA) identified 26 hub genes. Our analyses revealed eight prognostic genes, leading to establishing a six-gene model for assessing prognosis. Correlation analysis showed that the CAF risk score correlates with CAF infiltration and related markers. We also identified six potential drugs, observing significant differences between high-CAF and low-CAF risk groups. High CAF risk scores were associated with lower responses to immunotherapy and higher tumor mutation burdens. GSEA indicated that these scores are enriched in tumor microenvironment pathways. In summary, these six model genes can predict overall survival and responses to chemotherapy and immunotherapy for pancreatic cancer, offering valuable insights for future clinical strategies.

## 1. Introduction

Pancreatic cancer (PC) is one of the most aggressive and deadly cancers, with current treatment options largely ineffective. It ranks among the fifteen most common cancers globally, particularly affecting Western countries like Europe and North America [1]. PC is characterized by a desmoplastic reaction, leading to a dense, fibrous tumor stroma that creates high intertumoral pressure. This physical barrier hampers therapy by compressing blood vessels and contributing to chemoresistance [2]. The tumor microenvironment (TME) is crucial in cancer progression, affecting metastasis and treatment responses. It consists of various cells, extracellular matrix components, and signaling molecules that promote tumor growth and therapeutic resistance. Notably, cancer-associated fibroblasts (CAFs) comprise over 70% of the TME and actively interact with cancer cells and immune components. Understanding this complex relationship is essential for addressing challenges in cancer treatment [3,4,5]. Activated pancreatic stellate cells (PSCs) also contribute significantly to PDAC, creating a complex TME that promotes growth, invasiveness, and immunosuppression [6]. Single-cell RNA sequencing reveals that about 35% of cells in PDAC tumors are cancerous, while 26% are fibroblasts and PSCs. Despite similarities across studies, there is substantial variability in cellular distribution among patients [7,8,9,10,11]. This heterogeneity increases the likelihood of therapy failure in clinical trials and underscores the need for more sophisticated models during the in vitro research stage [12].

Cancer-associated fibroblasts (CAFs) are known to affect the TME through remodeling the extracellular matrix (ECM) and facilitating neoplastic cell invasion. Their interaction with cancer cells and other stromal components happens via secreting growth factors, cytokines, and chemokines, thereby shaping the TME and influencing tumor progression [13]. Identifying specific fibroblast markers has enabled the live cell sorting of CAF subpopulations, facilitating in-depth in vivo mechanistic studies [14]. Mechanistically, CAFs promote tumor migration through the mechanisms of contraction and proteolysis that remodel the ECM [13]. CAFs release substantial amounts of growth factors and proinflammatory cytokines, such as transforming growth factor-β (TGF-β), interleukin-6 (IL-6), and CC-chemokine ligand 2 (CCL2), which recruit immune cells, particularly immunosuppressive cells, into the tumor stroma, promoting immune evasion [15,16]. CAFs contribute to chemotherapy resistance through multiple mechanisms, such as CAFs upregulating glutathione levels in prostate cancer and inhibit reactive oxygen species (ROS) production [17], facilitating natural immunity to RAF inhibitors in cancers with BRAF mutations [18], expressing *CD10* and *GPR77* exhibit intrinsic resistance to chemotherapy [14], and regulating the NF-κB signaling pathway via secreting interleukin-6 (IL-6) and interleukin-8 (IL-8) in CD10^+^GPR77^+^ CAFs.

Weighted gene co-expression network analysis (WGCNA) [19] is a systems biology approach that directly identifies gene clusters (modules) based on their co-expression patterns across multiple samples [20]. This method facilitates the detection of highly correlated gene modules, summarizes them using module eigengenes or intramodular hub genes, and establishes relationships between modules and external sample traits through eigengene network analysis. Additionally, WGCNA allows for the assessment of module membership, providing insights into gene significance within specific biological contexts. It has been widely used to identify key biomarkers in various cancers. For instance, it has facilitated the discovery of Epstein–Barr virus (EBV) markers in gastric cancer [21], *CENPM* as a metastasis-related gene in adrenocortical carcinoma [22], and prognostic as well as diagnostic biomarkers for colorectal cancer [23]. It has also been applied to identify circulating microRNA signatures for the early detection of pancreaticobiliary cancer [24] and to explore the role of *HLF* in anaplastic thyroid cancer immunotherapy [25]. Similarly, CAF markers have been identified and utilized in multiple malignancies, including breast cancer [26], gastric cancer [27], intrahepatic cholangiocarcinoma [28], high-grade serous ovarian carcinoma [29], colorectal cancer progression [30], bladder cancer [31], and prostate cancer [27] among others.

This study aims to identify a gene signature associated with cancer-associated fibroblasts to enhance the prediction of prognosis and therapeutic responses in patients diagnosed with pancreatic cancer.

## 2. Results

### 2.1. Higher CAF Infiltrations Are Related to Worse Overall Survival in Pancreatic Cancer

The synopsis of the workflow is illustrated in Figure 1A. A total of 185 pancreatic cancer samples from TCGA-PAAD and 182 tumor tissues merged from GSE183795 and GSE78229 were analyzed, with 112 non-tumor and non-survival data samples being excluded. Four algorithms were evaluated: EPIC, xCell, MCPcounter, and the estimated algorithm. The results of the CAF infiltration and stromal score are shown in Appendix A (TCGA-PAAD) and Appendix A (GSE183795 and GSE78229). Subsequently, the Kaplan–Meier curve was utilized to analyze the association between high and low CAF scores with overall survival (OS), as depicted in (Figure 1B,C; Appendix A). The weighted gene co-expression network analysis (WGCNA) was performed based on the MCPcounter scores. The data from GSE183795 and GSE78229 and TCGA-PAAD were utilized, and genes were ranked according to their fold change. The top 5000 most differentially expressed genes (DEGs) were retained from the GEO dataset. The optimal soft threshold power (sft) was determined to be 8 (scale-free R2 = 0.996) (Figure 2A), while the TCGA exhibited a value of 6 (scale-free R2 = 0.972) (Figure 2B). The topology plot (GSE183795 and GSE78229) (Appendix A) and TCGA-PAAD (Appendix A) demonstrate the rationality of our subsequent clustering. Concurrently, the clustering of module eigengenes yielded a cut-off value < 0.2 in GSE183795 and GSE78229 (Figure 2C) and TCGA-PAAD (Figure 2D). The construction of the co-expression network [32] from the dendrogram-module color map reveals 19 co-expressions in GSE183795 and GSE78229 (Appendix A) and 20 co-expression modules in TCGA-PAAD (Appendix A). Module traits display that the strongest positive correlation with the CAF proportion (Cor = 0.92, *p* = 6 × 10^−73^ and stromal score (Cor = 0.91, *p* = 1 × 10^−71^) was the magenta module in GSE183795 and GSE78229 (Figure 2E). In contrast, the brown module exhibited the strongest positive correlation with the CAF proportion (Cor = 0.95, *p* = 1 × 10^−92^) and stromal score (Cor = 0.79, *p* = 8 × 10^−39^) was brown in TCGA-PAAD (Figure 2F). Following the identification of modules with the most significant relationship, magenta hub genes were selected from GSE183795 and GSE78229 (Appendix A) and brown (Appendix A) from TCGA-PAAD. The GEO-magenta module’s scatter plot results demonstrated a substantial correlation between gene significance (GS) and module membership (MM) in CAF (Cor = 0.93, *p* = 7.2 × 10^−106^) (Figure 2G) and stromal scores (Cor = 0.92, *p* = 3.3 × 10^−99^) (Figure 2H). A similar correlation was observed in the TCGA-brown module, MM and GS for CAF (Cor = 0.98, *p* < 1 × 10^−200^) (Figure 2I) and stromal scores (Cor = 0.89, *p* = 2.7 × 10^−171^) (Figure 2J) demonstrated a high degree of correlation. Utilizing gene Significance filter >0.4 and model Significance filter >0.8 as benchmark standards, 55 genes in the GSE183795 and GSE78229 black model and 195 in the TCGA-PAAD brown module were identified as hub genes, exhibiting a high correlation with CAF and stromal scores.

### 2.2. Intersection Genes Enrichment Related to ECM

The Venn diagram was used to visualize the intersection of two hub gene sets, which resulted in the identification of 26 genes (Figure 3A, Appendix A). Subsequently, gene ontology (GO) analysis was performed on these 26 genes. The analysis revealed that the regulation of neuron projection development exhibited significant enrichment in the biological process (BP) terms. The collagen-containing extracellular matrix and extracellular matrix structural constituents were identified as the primary enriched cell component (CC) and molecular function (MF) terms, respectively, as illustrated by the barplot (Appendix A) and bubble plot (Appendix A). Additionally, KEGG pathway enrichment analysis revealed significant enrichment in the cytoskeleton in muscle cells (Appendix A).

### 2.3. Establishing the Prognostic Risk Model Genes from the Stromal CAF-Based

A total of 185 cases from TCGA-PAAD were utilized as the training cohort due to the more substantial data available, while 182 pancreatic cancer samples from the GEO dataset (GSE183795 and GSE78229) were employed as the validation group. Eight core genes were screened from the 26 intersection genes, resulting in the creation of a favorable survival-related forest plot (Figure 3B). Subsequently, a univariate Cox regression on 26 shared hub genes, resulting in the identification of eight genes with an OS-related (*p*-value < 0.05). The coefficient profiles of these genes were then analyzed using the least absolute shrinkage and selection operator (LASSO) Cox regression (Appendix A). The adjustment factor (lambda), as depicted in Appendix A, was determined using partial likelihood variance through tenfold cross-validation. To create the CAF risk model, we identified six specific genes: the CAF risk score = *FSTL1* expression ∗ (−0.051) + *CTSK* expression ∗ (−0.93) + *PRRX1* expression ∗ 0.44 + *GFPT2* expression ∗ (−0.14) + *MMP2* expression ∗ 0.68 + *MFAP5* expression ∗ 0.13 (Appendix A).

### 2.4. High CAF Infiltrations and Markers Can Be Shown from CAF Signature Genes

To enhance the reliability of the CAF model as a forecasting tool, Spearman’s correlation analysis was conducted to establish a relationship between CAF risk and stromal score, as well as CAF abundances, as indicated by EPIC, alongside xCell, MCP-counter, and TIDE. It was observed that there was a positive correlation between the CAF risk score and the multi-estimated CAF infiltrations along with the stromal score in both GSE183795 and GSE78229 (Figure 4A) and TCGA-PAAD (Figure 4B) groups. Furthermore, the heatmap, which revealed the expression patterns of CAF markers (Appendix A), identified six CAF module genes associated with the CAF risk score in GSE183795 and GSE78229 (Figure 4C) and TCGA-PAAD (Figure 4D) cohorts. Additionally, we observed a high and positive correlation between the CAF risk score and the expression levels of the two genes and the collected CAF markers in both GSE183795 and GSE78229 (Figure 4E) and TCGA-PAAD (Figure 4F) cohorts.

### 2.5. Chemo-Immuno-Therapy Sensitivity Prediction and Molecular Docking

Pancreatic cancer is recognized as the leading cancer within the digestive system. Surgical intervention often yields suboptimal outcomes, underscoring the necessity for adjuvant treatment modalities, such as chemotherapy and immunotherapy, to enhance clinical outcomes. In this study, we employed the GDSC online database (accessed on 2 March 2025) to screen chemotherapy drugs based on their association with CAF-risk groups. This analysis identified 13 sensitivity drugs (Appendix A) that satisfied the *p* < 0.001 in high-/low-risk groups in TCGA-PAAD and 30 in GSE183795 and GSE78229 (Appendix A). Six intersection drugs, including staurosporine, dasatinib, OTX015, BMS-536924, Luminespib, and IGF1R, were shown in both the GSE183795 and GSE78229 (Figure 5A) and TCGA-PAAD cohorts (Figure 5B). Immunotherapy using immune checkpoint inhibitors has generated optimism among PC patients. To assess whether the CAF risk score could serve as an immunotherapy predictor for pancreatic cancer patients, we employed the TIDE method. In Figure 5C, bar charts showed that the low-CAF group exhibited a higher immunotherapy response (58%) than the higher-CAF group (30%), which was significant (*p* < 0.001) in GSE183795 and GSE78229. The low-CAF group exhibited a higher immune-therapy response (58%) than the higher-CAF group (21%), which is significant (*p* < 0.001) in TCGA-PAAD. In Figure 5D, violin plots illustrate that the low-risk group receiving a TIDE score, and the high-risk group receiving a higher score between GSE183795, GSE78229, and TCGA-PAAD. Additionally, the AUC figures of 0.725 (95%CI: 0.650–0.797) in GSE183795 and GSE78229 (Figure 5E). Furthermore, the AUC figures of 0.773 (95% CI: 0.696–0.844) in TCGA-PAAD (Figure 5F) demonstrated the superior efficacy of our CAF model in forecasting immunotherapy responses. The 3D structure of the chemical drugs was obtained from PubChem (https://pubchem.ncbi.nlm.nih.gov/) (accessed on 2 March 2025), and the 3D structure of the module genes was obtained from the Protein Data Bank (https://www.rcsb.org/) (accessed on 2 March 2025). Utilizing the CB-DOCK2 online platform (http://cadd.labshare.cn/cb-dock2/index.php) (accessed on 2 March 2025). The Vina score of these interactions was found below (Table 1), indicating robust binding characteristics. We show their 3D binding spatial structure of the least binding energies between the gene and drugs, including MMP2 with OTX015 (Figure 6A), FSTL1 with OTX015 (Figure 6B), GFPT2 with Luminespib (Figure 6C), and CTSK with staurosporine (Figure 6D). The potential drugs for expanding the drug repertoire for pancreatic cancer are enhanced by these findings.

### 2.6. High CAF Risk Is Correlated with Tumor Mutation Burden

After this observation, it was noted that there is a discrepancy in tumor mutation burden between the CAF high-risk group (83.5%, Appendix A) and low-risk group (81.7%, Appendix A). The relationship between high- and low-risk groups and TMB is demonstrated in Appendix A. Spearman’s correlation analysis revealed a relationship between CAF-risk score and TMB in Appendix A. The correlation between different score algorithms and TMB is illustrated in Appendix A.

### 2.7. GSEA and ssGSEA of the Six Genes of the CAF Signature

Gene Set Enrichment Analysis (GSEA) was utilized to compare high and low CAF-infiltration pancreatic cancer in GSE183795 and GSE78229 cohorts, as well as the TCGA-PAAD cohorts. The high-risk group in GEO datasets (Figure 7A) and TCGA datasets (Appendix A) exhibited significant enrichment of genes belonging to the extracellular matrix (ECM) structural constituent category, along with a concomitant association with tumor evasion and immigration. This finding underscores the pivotal role of CAFs in modulating the tumor microenvironment. In contrast, the low-risk group in GEO datasets (Figure 7B) and TCGA datasets (Appendix A) demonstrated a predominant enrichment in cell metabolism and signal transduction pathways, which are more intimately linked to basic cellular function. Furthermore, single-sample gene set enrichment analysis (ssGSEA) revealed consistent enrichment of five Gene Ontology (GO) biological processes: bone morphogenesis (GEO datasets in Figure 7C, TCGA dataset in Appendix A), cell fate specification involved in pattern specification (GEO datasets in Figure 7D, TCGA dataset in Appendix A), embryonic neurocranium morphogenesis (GEO datasets in Figure 7E, TCGA dataset in Appendix A), and fibroblast growth factor receptor signaling pathway (GEO datasets in Figure 7F, TCGA dataset in Appendix A), mesenchyme development (GEO datasets in Figure 7G, TCGA dataset in Appendix A). The correlation coefficients (R^2^) for both datasets ranged over 0.5, and the *p* < 2 × 10^−13^ GSE183795 and GSE78229, and TCGA-PAAD. These findings further support the valuable role of CAFs in ECM components and tumor invasion and immigration in pancreatic cancer. In a similar vein, the KEGG pathway analysis revealed an enrichment of cytokine receptor and extracellular matrix reshape and invasion in the high-risk group in GSE183795 and GSE78229 (Appendix A) and TCGA-PAAD (Appendix A). In contrast, the low-risk groups exhibited a significant association with basic cell biophysical processes in GSE183795 and GSE78229 (Appendix A) and TCGA-PAAD (Appendix A).

### 2.8. Validation of Key Genes in CCLE and HPA Databases

Utilizing the CCLE database, it was confirmed that fibroblast cell lines exhibited elevated mRNA levels of the six module genes (GFPT2, MFAP5, CTSK, MMP2, FSTL1, and PRRX1) (accessed on 2 March 2025) compared to pancreatic cancer cell lines (Figure 8A). The Wilcoxon analysis (Figure 8B) shows the FSTL1 (*p* < 2.22 × 10^−16^), CTSK (*p* < 2.22 × 10^−16^), PRRX1 (*p* < 2.22 × 10^−16^), GFPT2 (*p* = 0.00015), MMP2 (*p* < 2.22 × 10^−16^), and MFAP5 (4.1 × 10^−13^) genes exhibited higher expression in fibroblasts than in the pancreas. To further substantiate the protein expression traits of six CAF signature genes, the HPA database’s (accessed on 2 March 2025) IHC images were obtained to illustrate higher expression in CAFs than in pancreatic cancer (Figure 8C). These confirmations suggested the possibility of these genes being markers specific to CAFs.

## 3. Discussion

Cancer-associated fibroblasts (CAFs), as the predominant component of the tumor microenvironment (TME), play a vital role in pancreatic cancer, influencing tumor initiation, metastasis, drug sensitivity, immune evasion, and prognosis [33]. Similarly, our research found that a high CAF score was associated with poor overall survival. To the best of our knowledge, our research is pioneering in methodically disclosing the function of CAFs in pancreatic cancer through WGCNA co-expression analysis. As illustrated in the workflow (Figure 1A), two cohorts were analyzed: GSE183795 and GSE78229 from GEO and TCGA-PAAD from TCGA. Utilizing sophisticated computational algorithms, including WGCNA, univariate Cox regression, Spearman’s correlation, and LASSO regression, we identified six targeted genes (GFPT2, MFAP5, CTSK, MMP2, FSTL1, and PRRX1) that exhibited a strong overall survival (HR > 1). These findings substantiate the validity and robustness of the CAF-associated model genes in pancreatic cancer. The GSDC database (accessed on 2 March 2025) was then queried to identify six intersection drugs that exhibited higher sensitivity in the high-risk groups compared to the low-risk group. The 3D structures of four gene proteins and five drugs were obtained from a designated website. Subsequent molecular docking analyses revealed that binding of these drugs was highly stable, as evidenced by their binding structures and energies. Notably, GFPT2 demonstrated the lowest binding energy of −9.1 kcal/mol when bound to Luminespib, which may be a potential drug for high GFPT2 expression PC patients. The application of the online TIDE algorithm revealed that the group with higher TIDE scores exhibited an increased rate of non-responsiveness to immunotherapy in both the GSE183795 and GSE78229 and TCGA-PAAD groups. The ROC analysis further validated the robustness of our findings.

In the high CAF-infiltration group, GSEA revealed enrichment in extracellular matrix (ECM) structural constituents and related to tumor evasion and immigration, thereby highlighting the role of CAFs in modulating the tumor microenvironment. Conversely, the low CAF-infiltration group exhibited enrichment in cell metabolism and signal transduction, which are more connected to basic function. Furthermore, the ssGSEA findings revealed a direct relationship between epithelial–mesenchymal transition (EMT) and ECM, which plays a part in the tumor microenvironment in pancreatic cancer. CAFs reshape and secrete the ECM [34]. CAF-mediated collagen-induced ECM alterations depend not solely on the binding and activating discoidin domain receptors (DDRs) [35]. The enhanced expression of DDR1 and DDR2 collagen receptors in melanoma cells was observed when grown on 3D decellularized ECM from melanoma-related CAFs, in contrast to matrices from regular dermal fibroblasts, highlighting the tumor-enhancing function of CAF-derived ECM [36]. In addition to the receptor activation, CAFs secrete matrix-remodeling enzymes, such as lysyl oxidase (LOX), which induce ECM stiffness, thereby promoting cancer cell proliferation, invasion, and epithelial-to-mesenchymal transition (EMT) in oral squamous cell carcinoma (SCC) in vitro [37] and facilitating gastric cancer-derived liver metastasis in vivo [38]. Furthermore, CAFs regulate the mechanical properties of ECM, including mechanical forces [39], stiffness [40], rigidity [41], shear stress, and viscosity [42,43]. These biomechanical changes can further drive tumor progression and therapy resistance. Additionally, CAF-derived ECM contributes to macrophage infiltration, influencing tumor matrix remodeling and poor prognosis in ovarian cancer patients [44]. Furthermore, CAF-derived matrix proteins have been associated with a desmoplastic ECM signature, which is characteristic of aggressive tumor phenotypes [45]. In addition, the impact of CAFs and ECM components on dendritic cell (DC)-mediated immune responses in various cancers [46]. These components can regulate DC phenotypes, thereby inducing immune tolerance and consequently, facilitating immune evasion [47].

We assessed CAF infiltration in pancreatic cancer to ensure robustness and prevent overfitting. The EPIC method built the model, while the xCell, MCP-counter, and TIDE methods validated it. Our study confirms the established role of CAFs as crucial components of the tumor microenvironment (TME), influencing various aspects of tumor behavior, such as initiation, metastasis, and immune evasion. Similar studies, like those by Kaluri et.al. [48] have highlighted the importance of CAFs in tumor progression across different cancers, reinforcing our observations. The CCLE dataset revealed elevated expression of six module genes (GFPT2, MFAP5, CTSK, MMP2, FSTL1, and PRRX1) associated with poor overall survival in patients with pancreatic cancer. This aligns with findings from other studies that have also discovered specific gene signatures linked to CAF activity and patient prognosis. For instance, a recent study by Yu et al., [49] also demonstrated CAF gene signatures in pancreatic cancer correlating with survival outcomes.

MMP2, a matrix metalloproteinase, is a promising early marker for PDAC. It plays a role in tissue remodeling, inflammation, and cancer progression [50,51,52,53,54,55,56,57,58,59,60,61,62].

PRRX1, a member of the paired-related homeobox family of transcription factors, has been implicated in the development of acinar-ductal metaplasia (ADM) and regeneration. Additionally, it impacts gemcitabine sensitivity via its role in epithelial–mesenchymal transition (EMT) [63,64,65]. Our findings confirm that PRRX1 is more expressed in CAFs than in normal pancreatic tissue, and its suppression may improve chemotherapy effectiveness.

MFAP5 is associated with poor prognosis in cancers like ovarian and intrahepatic cholangiocarcinoma, and high MFAP5 expression in CAFs correlates with unfavorable outcomes in pancreatic cancer [66,67].

We utilized molecular docking to evaluate drug interactions with target proteins, particularly highlighting GFPT2’s significant binding with Luminespib as a potential therapeutic option. This technique has been similarly employed in studies such as that by Chen et al. [68] underlining the utility of computational methods in understanding drug efficacy associated with specific biomarkers. Our method of assessing CAF infiltration using tools such as EPIC and xCell, corroborated by the TIDE method, is further supported by methodologies in other studies that also aim to validate CAF-related models, such as Zhu et al. [68], reinforcing the rigor of our analytical approach.

In summary, while our research contributes novel insights into the functions of CAFs in pancreatic cancer and establishes a specific gene signature, it also aligns well with the existing literature, highlighting a growing consensus on the multifaceted roles of CAFs in tumor biology and therapy resistance. These comparisons not only validate our findings but also enhance the understanding of CAFs in the context of cancer research as a whole.

Limitations: Despite the confidence in the functionality of pancreatic cancer genes and the docking of target genes with drug molecules, several limitations must be acknowledged. Firstly, the present bioinformatic analysis is retrospective, drawn from two publicly available gene expression datasets, and lacking multi-center data. Second, there is a deficiency in real-world study data concerning CAFs in pancreatic cancer, and the CAF marker has not been verified at the molecular, cellular, and animal levels. Nonetheless, our study offers novel insights into the relationship between CAF infiltration and pancreatic cancer.

## 4. Materials and Methods

### 4.1. Accumulating and Processing Data

Data on RNA sequencing and clinical traits of patients with pancreatic cancer were obtained from two groups for thorough examination. The fragments per kilobase of transcript per million mapped reads (FPKM) formatted RNA-seq data and corresponding clinical prognostic records (including overall survival time and vital status) for 185 pancreatic adenocarcinoma (TCGA-PAAD) samples were retrieved from The Cancer Genome Atlas (TCGA) Genomic Data Commons (GDC) portal (https://portal.gdc.cancer.gov/ (accessed on 2 March 2025), with data freeze date on 10 February 2025. Tumor mutation burden (TMB) data were simultaneously acquired through the TCGA Mutation Annotation Format (MAF) files. Normalized FPKM values were converted to transcripts per million (TPM) using the following formula:TPM = FPKM × 10^6^/∑FPKM

Followed by log2 (TPM + 1) transformation to approximate normality [69,70]. We extracted the normalized expression matrix and clinical metadata of 295 pancreatic cancer samples (Accession: GSE183795 and GSE78229) [71,72] were downloaded from the Gene Expression Omnibus (GEO) repository (https://www.ncbi.nlm.nih.gov/geo/) accessed on 2 March 2025. Samples with >20% missing expression values or no survival data were excluded for both cohorts. Finally, we included 185 samples from TCGA-PAAD and 182 samples from GSE183795 and GSE78229. There is no ethical issue because our study utilized the publicly available datasets online.

### 4.2. Quantification of Cancer-Associated Fibroblasts (CAFs)

To comprehensively characterize CAF infiltration within the tumor microenvironment (TME), we integrated four established computational algorithms complemented by stromal scoring analysis: EPIC (Estimating the Proportion of Immune and Cancer cells) [73], xCell [74], MCP-counter (Microenvironment Cell Populations-counter) [75], and TIDE (Tumor Immune Dysfunction and Exclusion) [76,77]. The four algorithms EPIC (v 1.1.7), (https://github.com/GfellerLab/EPIC, accessed on 2 March 2025), xCell (v 1.1.0) (https://github.com/dviraran/xCell, accessed on 2 March 2025), and MCP-counter (v 1.2.0) (https://github.com/ebecht/MCPcounter, accessed on 2 March 2025) analyses were based on the “limma ()” R package (version “4.4”) [78] with default parameters. TIDE scores were obtained through the official web portal (http://tide.dfci.harvard.edu/, accessed on 2 March 2025) using pre-processed log2 (TPM + 1) expression matrices. The ESTIMATE algorithm [79] was additionally applied via the estimate R package (v 1.0.13) to calculate stromal scores reflecting global stromal infiltration levels. This stromal index served as orthogonal validation for CAF quantification. We normalized the different methods of CAF scores to generate a consensus CAF infiltration index and reduce algorithm biases.

### 4.3. CAF and Stromal Co-Expression Network Module

The WGCNA R package (v 1.73) was utilized to conduct a weighted gene co-expression network analysis (WGCNA) [19], aiming to build co-expression networks and pinpoint key genes linked to the infiltration of cancer-related fibroblasts (CAF) and their stromal ratings. WGCNA identifies gene modules based on similarity in expression patterns and detects hub genes highly connected within relevant modules. Gene expression profiles from the TCGA-PAAD and GSE183795 and GSE78229 cohorts were analyzed. The leading 5000 genes exhibiting the most significant median absolute deviation (MAD) were chosen to construct the network. A Pearson correlation similarity matrix was computed and transformed using a soft-thresholding power to approximate scale-free topology. The adjacency matrix was converted into a topological overlap matrix (TOM), followed by hierarchical clustering based on TOM-derived dissimilarity. The identification of gene modules was achieved through the dynamic tree-cut algorithm, requiring a minimum of 30 genes, and selected 1000 genes for visualization. The initial principal component of each module, known as module eigengenes (MEs), showed a correlation with CAF infiltration levels and stromal scores estimated by EPIC. The most relevant module was selected for further analysis. Core genes were identified based on gene significance (GS > 0.4) and module membership (MM > 0.8), indicating a strong association with the trait and module. Intersection hub genes were considered as our final hub genes in these two cohorts.

### 4.4. Kaplan–Meier Curve and Log-Rank Analysis

Cohorts included from the GSE183795, GSE78229, and TCGA-PAAD were grouped into high-/low-CAF scores, and overall survival (OS) was determined using Kaplan–Meier curves and checked by log-rank analysis based on a scoring algorithm. Subsequently, we also verified our constructed model genes.

### 4.5. Gene Ontology (GO) and the Kyoto Encyclopedia of Genes and Genomes (KEGG) Analysis

The enrichment analysis of Gene Ontology (GO) and KEGG pathways was performed to delineate the biological importance of the pinpointed central genes. Functional categories, including biological processes (BPs), molecular functions (MFs), and cellular components (CCs), were analyzed using the “clusterProfiler” R package (version 3.20) [80,81]. *p* < 0.05 was considered statistically significant. The hub genes enrichment results were visualized using bar and bubble plots.

### 4.6. CAF-Associated Model Construction and Validation

To further explore the biological implications of CAF-related hub genes, a CAF-associated model was developed based on the TCGA-PAAD cohort, along with the GSE183795 and GSE78229 datasets for validation. Univariate analysis was performed to identify genes significantly correlated with CAF infiltration. Significant genes were subjected to the least absolute shrinkage and selection operator (LASSO) regression with 1000 iterations to refine the model, using the “glmnet” R package (version 4.1.8) [82]. The CAF-related model was constructed using the formula: CAF score = ∑ (β_i ∗ Exp_i), where β_i represents the LASSO coefficient of the i-th gene, and Exp_i denotes its expression value. Patients were stratified into high- and low-CAF-score groups based on the median score, and differences in CAF-associated characteristics between the groups were analyzed. The model`s robustness was validated in the TCGA-PAAD cohort to ensure reproducibility and biological relevance.

### 4.7. Sensitivity Drugs Prediction and Molecular Docking

To evaluate the potential clinical implications of the CAF-associated model, drug sensitivity and immunotherapy response predictions were conducted. The chemotherapeutic response was estimated based on the Genomics of Drug Sensitivity in Cancer (GDSC) database (https://www.cancerrxgene.org/) (accessed on 2 March 2025) [83]. Additionally, the Tumour Immune Dysfunction and Exclusion (TIDE) algorithm (http://tide.dfci.harvard.edu/) (accessed on 2 March 2025) was employed to predict responses to immune checkpoint blockade (ICB) therapy [76,77]. The chi-squared test was used to evaluate the variance in reaction rates among groups with high and low CAF scores. Subsequently, an assessment of the CAF-associated model’s forecasting accuracy was conducted using receiver operating characteristic (ROC) curves and their respective area under the curve (AUC) values. This analysis provided insight into the model’s potential effectiveness in directing tailored treatment approaches. Then, we employed the module genes with sensitive drugs to binding, which conducted molecular docking to assess the binding energies and interaction patterns between candidate drugs and their targets. Sensitivity drugs’ structural data were sourced from the PubChem Compound Database (https://pubchem.ncbi.nlm.nih.gov/) (accessed on 2 March 2025) [84] and downloaded in SDF format. Potential protein structural data were downloaded from the Protein Data Bank (https://www.rcsb.org/) (accessed on 2 March 2025) in PDB format. The top essential drugs and the proteins encoded by the respective target genes were subjected to molecular docking using the computerized Protein Ligand Docking online tools (http://cadd.labshare.cn/cb-dock2/index.php) (accessed on 2 March 2025) [85]. The interaction graph was downloaded, and the Vina scores were saved. By identifying ligands that exhibit high binding affinity and beneficial interaction patterns, we can prioritize drug targets for additional experimental validation and refine the design of prospective candidate drugs.

### 4.8. Gathering and Analyzing Data on Somatic Changes

Somatic mutation data for the TCGA-PAAD cohort were obtained from the TCGA GDC (https://portal.gdc.cancer.gov/) (accessed on 2 March 2025). The 20 most frequently mutated genes in high- and low-CAF-score groups were identified and visualized using the “maftools” R package (3.20) [86]. The tumor mutation burden (TMB) is recognized as a potential indicator of the efficacy of immuno-therapy [87], and it was calculated for each PAAD sample using the “tmb” function in the “maftools” package. The relationship between TMB and CAF risk scores was assessed using Spearman’s correlation analysis to explore potential associations between mutational load and CAF-related tumor characteristics.

### 4.9. GSEA and ssGSEA Enrichment Analyses

A gene set enrichment analysis (GSEA) was conducted to identify key and KEGG pathway gene clusters that exhibited variation in enrichment between groups with high and low CAF scores in the GSE183795 and GSE78229 and TCGA-PAAD cohorts. This analysis was conducted using the enrichplot and “clusterProfiler” R packages, utilizing gene sets “c2.cp.kegg.Hs.symbols.gmt” and “c5.go.Hs.symbols.gmt” from the Molecular Signatures Database (MSigDB) [88]. Additionally, single-sample GSEA (ssGSEA, R package version 3.20) was applied to quantify the enrichment scores of hallmark gene sets related to dilated cardiomyopathy, focal adhesion, hypertrophic cardiomyopathy HCM, and TGF-beta signaling pathway [89]. Spearman’s correlation analysis evaluated associations between CAF risk scores and gene set enrichment scores, providing insights into the functional pathways linked to CAF activity.

### 4.10. Validation Via Cancer Cell Line Encyclopedia (CCLE) and Human Protein Atlas (HPA) Databases

To validate the findings at the cellular level, the mRNA expression data of the identified markers in 37 fibroblasts and 41 pancreatic cell lines were retrieved from the Cancer Cell Line Encyclopedia (CCLE) database (https://portals.broadinstitute.org/ccle) (accessed on 2 March 2025) [90]. Differential expressions between fibroblasts and GC cell lines were analyzed using Wilcoxon tests and visualized with heat maps. At the protein level, immunohistochemical (IHC) staining images of these markers in pancreatic cancer tissues were obtained from the Human Protein Atlas (HPA) database (https://www.proteinatlas.org/) (accessed on 2 March 2025) [91]. Protein localization and expression patterns in pancreatic cancer tissues were visually inspected to corroborate the findings further.

### 4.11. Statistical Analysis

All statistical evaluations were conducted using R software (version 4.4.2, available on 31 October 2024; https://www.r-project.org/) (accessed on 2 March 2025). Pancreatic cancer patients were categorized into high and low-CAF-risk categories based on the median CAF risk score, which served as the threshold value. The Wilcoxon rank-sum test was employed to conduct paired analyses within each group. The Kaplan–Meier survival curves were employed to evaluate survival variances among sub-groups, and the log-rank test was used to determine statistical significance through the “survival” (v 3.8.3) and “survminer” (v 0.5.0) R packages. A *p* < 0.05 was deemed to hold statistical significance. Correlation analyses utilized Spearman’s rank correlation test. A multivariate Cox cregression analysis was conducted to identify distinct prognostic elements, and the CAF risk score’s forecasting precision was assessed through receiver operating characteristic (ROC) curves and the area under the curve (AUC) metrics. The generation of data visualization, encompassing heatmaps, scatter plots, and survival curves, was facilitated by R software, GraphPad Prism (version 8.4.0, launched on 20 February 2020), Adobe Photoshop (version 2024), and Microsoft Office 365 to ensure efficient management of data, comprehensive statistical evaluation, and the generation of figures.

## 5. Conclusions

In conclusion, our study successfully identified a module of six key genes—GFPT2, MFAP5, CTSK, MMP2, FSTL1, and PRRX1—through Weighted Gene Co-expression Network Analysis (WGCNA) to assess cancer-associated fibroblast (CAF) infiltration in pancreatic cancer (PC). The robustness of these module genes highlights their potential as significant biomarkers for evaluating prognosis and predicting responses to both chemotherapy and immunotherapy. The findings underscore the critical role of the tumor microenvironment (TME) in pancreatic cancer progression and treatment efficacy. Our results suggest that further exploration of the TME’s interactions and the implications of CAF infiltration might yield valuable insights into the clinical management of PC. This opens avenues for future research aimed at optimizing therapeutic strategies and enhancing patient outcomes through targeted approaches that consider CAF-related dynamics within the TME. Ultimately, a deeper understanding of these molecular markers and their pathways could lead to more personalized and effective treatment protocols in the fight against pancreatic cancer.

## Figures and Tables

**Figure 1 ijms-26-04876-f001:**
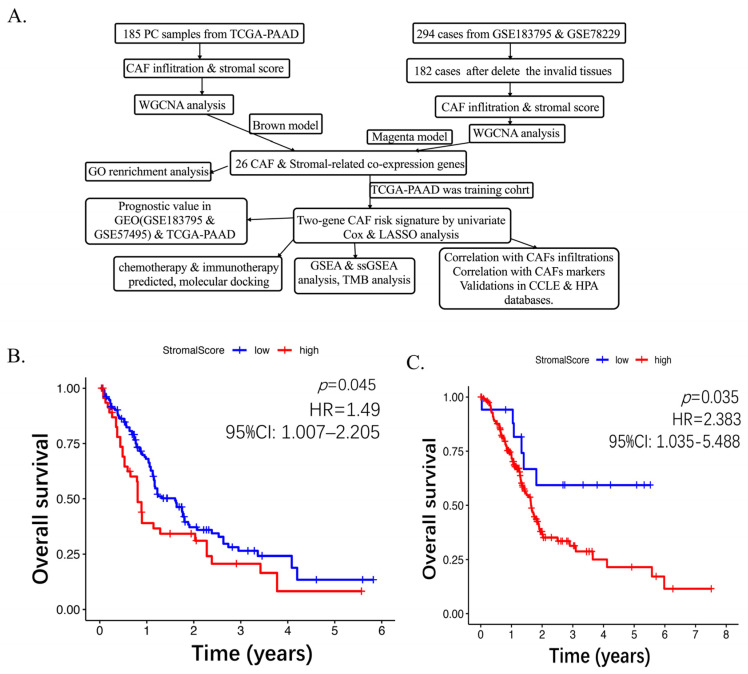
Workflow of the research on TCGA-PAAD and GSE183795 and GSE78229 (**A**). Overall survival analysis by log-rank. K-M curves show the pancreatic cancer patients with higher stromal-score related to bad overall survival in GSE183795 and GSE78229 (**B**) and TCGA-PAAD (**C**).

**Figure 2 ijms-26-04876-f002:**
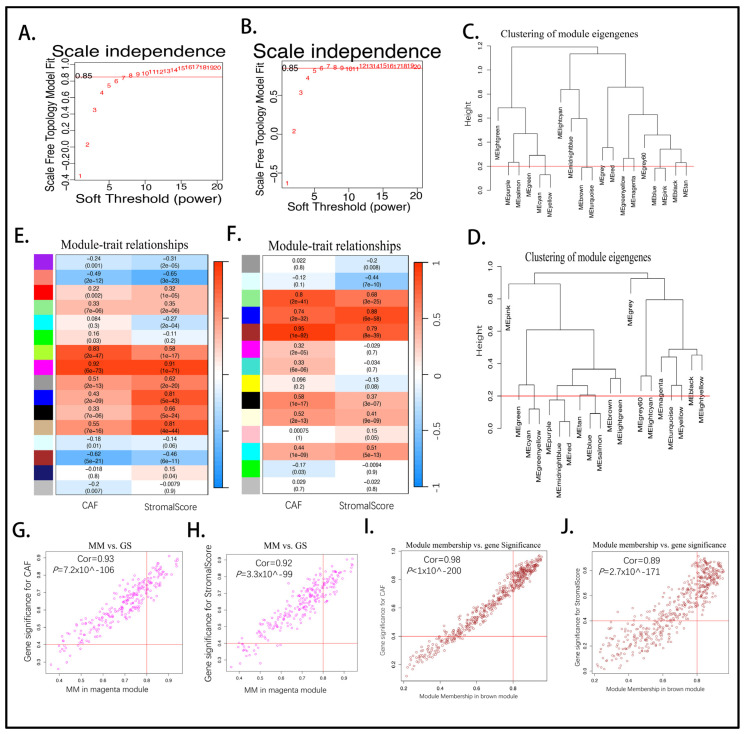
Co-express network of the WGCNA. Based on the scale topology tree in GSE183795 and GSE78229 (**A**) and TCGA-PAAD (**B**), the soft threshold power was decided. The cluster gene tree shows the depth of cut needed <0.2 displayed in GSE183795 and GSE78229 (**C**) and TCGA-PAAD (**D**). Module-trait relationships show different module colors’ relation values in GSE183795 and GSE78229 (**E**) and TCGA-PAAD (**F**). Scatter graphs of the module membership and gene significance of each gene in the magenta module in GSE183795 and GSE78229 for CAF (**G**) and for StromalScore (**H**); the brown module in brown TCGA-PAAD for CAF (**I**) and for StromalScore (**J**) (MM: module membership, GS: gene significance). The horizontal axis is the correlation between gene and co-expression modules, and the vertical axis is about the gene and phenotype.

**Figure 3 ijms-26-04876-f003:**
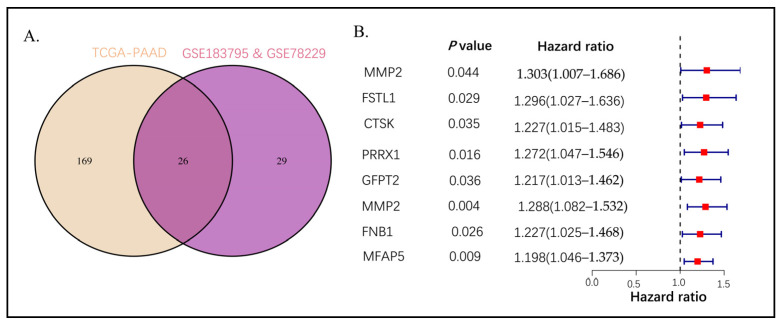
(**A**) The Venn diagram presented the intersection of TCGA-PAAD brown and GSE183795 and GSE78229 magenta module genes. (**B**) Univariate Cox analysis for the screening of overall survival-associated genes in TCGA-PAAD.

**Figure 4 ijms-26-04876-f004:**
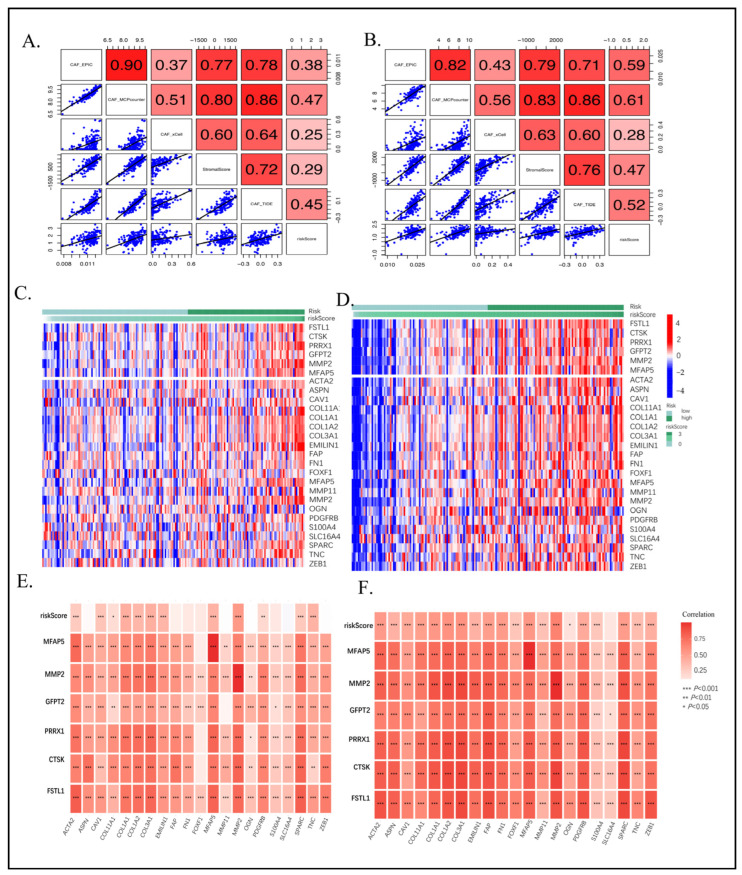
Spearman’s correlation analyses revealed the CAF risk score was strongly and positively correlated with stromal scores and multi-estimated CAF infiltrations in GSE183795 and GSE78229 (**A**) and TCGA-PAAD (**B**) cohorts. The heatmap revealing the expression patterns of CAF markers identified two CAF genes with the CAF risk score in GSE183795 and GSE78229 (**C**) and TCGA-PAAD (**D**) cohorts. The CAF risk score and two signature genes were positively correlated with the literature that reported CAF markers in GSE183795 and GSE78229 (**E**) and TCGA-PAAD (**F**) cohorts. (* means *p* < 0.05, ** means *p* < 0.01, *** means *p* < 0.001).

**Figure 5 ijms-26-04876-f005:**
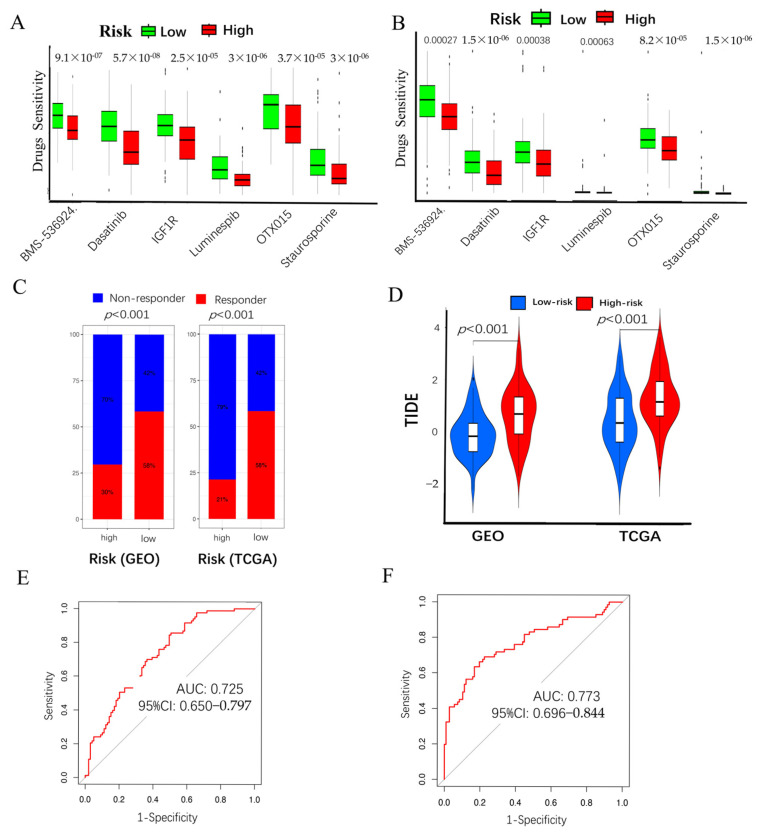
Sensitivity drugs: about the GSE183795 and GSE78229 (**A**) and TCGA-PAAD (**B**) including Staurosporine, Dasatinib, OTX015, BMS-536924, Luminespib, IGF1R. Immunotherapy responses of the high-/low-risk groups based on TIDE scores. The box plots (**C**) in GSE183795 and GSE78229 and TCGA-PAAD, violin plots (**D**) in GSE183795 and GSE78229 and TCGA-PAAD, and the curve showing the sensitivity of different methods in GSE183795 and GSE78229 (**E**) and TCGA-PAAD (**F**).

**Figure 6 ijms-26-04876-f006:**
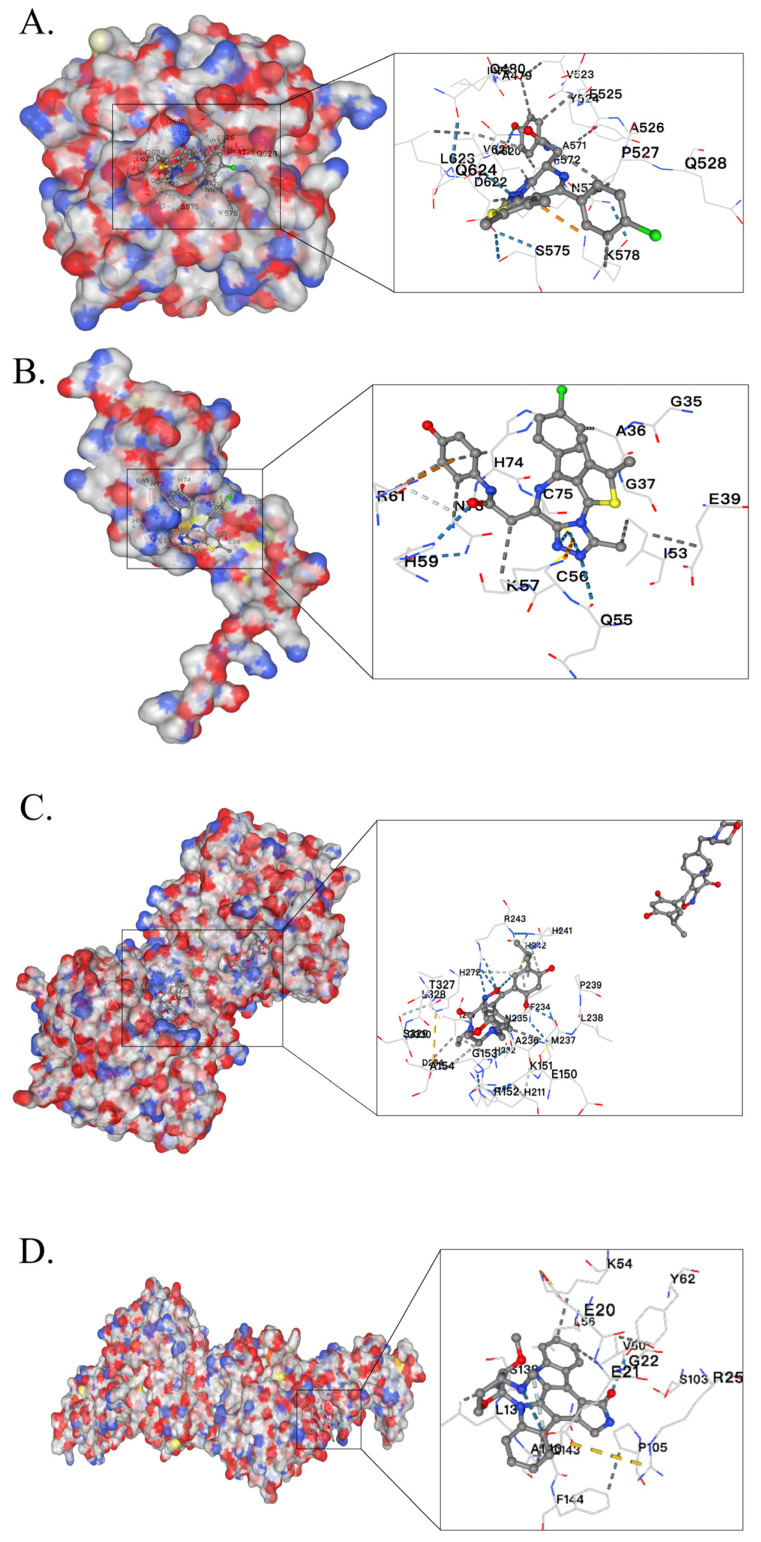
Molecular docking graphs MMP2 binding with OTX015 (**A**), FSTL1 with OTX015 (**B**), GFPT2 with Luminespib (**C**), and CTSK with Staurosporine (**D**).

**Figure 7 ijms-26-04876-f007:**
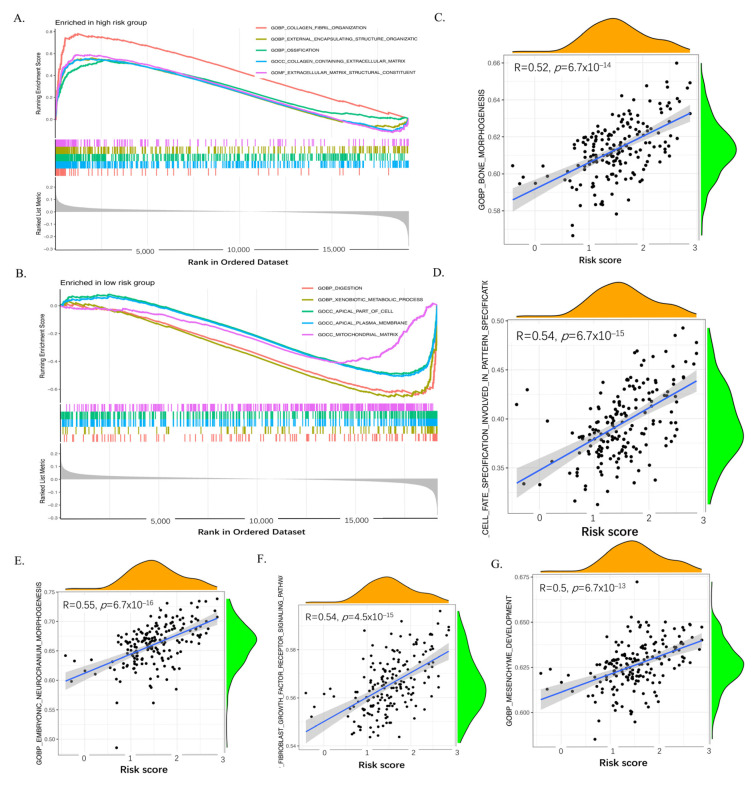
GSEA and ssGSEA enriched GO biological function in GEO datasets. Gene set enrichment analysis (GSEA) sets between high-CAF risk groups (**A**) and low risk group (**B**) in GSE183795 and GSE78229. ssGSEA results showed that the CAF risk score was positively correlated with bone morphogenesis (**C**), cell fate specification involved in pattern (**D**), embryonic neurocranium morphogenesis (**E**), fibroblast growth factor receptor signaling pathway (**F**), and mesenchyme development (**G**) in GSE183795 and GSE78229.

**Figure 8 ijms-26-04876-f008:**
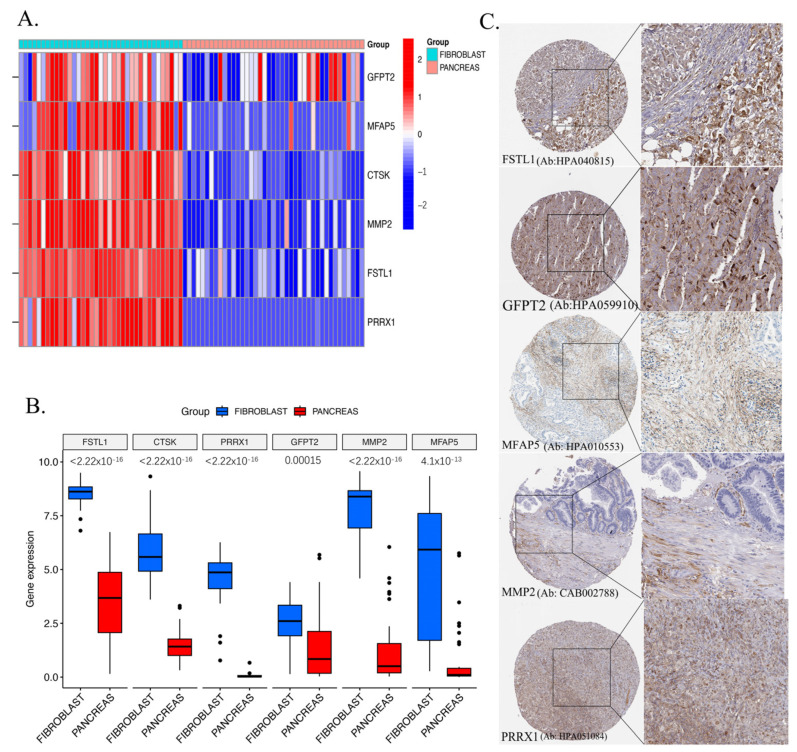
The mRNA expression levels of the two CAF genes in the fibroblasts and pancreatic cancer cell lines were illustrated in the heat map (**A**) and compared by Wilcoxon analysis (**B**). Protein expression in pancreatic cancer tissues (**C**).

**Table 1 ijms-26-04876-t001:** Molecular docking result of available protein and drugs.

Target	PDB ID	Drug	PubChem ID	Binding Energy (kcal/mol)
MMP2	1RTG	Staurosporine	44259	−7
		Dasatinib	3062316	−8.2
		OTX015	9936746	−8.4
		BMS-536924	135440466	−8
		Luminespib	135539077	−8.1
FSTL1	6JZA	Staurosporine	44259	−7.4
		Dasatinib	3062316	−7.3
		OTX015	9936746	−7.6
		BMS-536924	135440466	−7
		Luminespib	135539077	−7
GFPT2	7NUT	Staurosporine	44259	−8
		Dasatinib	3062316	−8.5
		OTX015	9936746	−8.5
		BMS-536924	135440466	−9
		Luminespib	135539077	−9.1
CTSK	8V57	Staurosporine	44259	−9
		Dasatinib	3062316	−7.6
		OTX015	9936746	−8.5
		BMS-536924	135440466	−7.9
		Luminespib	135539077	−7.5

## Data Availability

Data are contained within the article and Appendix A. The data presented in this study are available on request from the corresponding author.

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
