# Peer review of "Identification of Cancer Associated Fibroblasts Related Genes Signature to Facilitate Improved Prediction of Prognosis and Responses to Therapy in Patients with Pancreatic Cancer"

_ijms, 2025, doi:10.3390/ijms26104876_

Round 1
Reviewer 1 Report
Comments and Suggestions for Authors
Comments and Suggestions:
Title: Identification of Cancer associated fibroblasts related genes signature to facilitate improved prediction of prognosis and responses to therapy in patients with pancreatic cancer
Reviewer’s report:
The manuscript by Zhou et al., is very well written in an explained manner. The study emphasized to identify cancer associated fibroblasts (CAFs) related genes in pancreatic cancer (PC) using TCGA and GEO databases via WGCNA approach and identified 26 intersection hub genes. out of which 6 genes GFPT2, MFAP5, CTSK, MMP2, FSTL1, and PRRX1 were established for prognostic assessment and 6 drugs were screened; high-/low-CAF risk groups showed big differences. GSE analysis also showed the enrichment of high CAF risk scores in tumor micro-environment pathways. They concluded that the findings provide new insight and may offer a potential treatment option for chemotherapy and immunotherapy strategies in PC.
The manuscript is very well compiled. But still few points need to be addressed.
- Figure 2A and 2B: The cutoff seems to be 11 for TCGA-PAAD dataset. But the authors mentioned 8 in line 135. Similarly, in figure 2B, it seems there is no soft threshold power above the cutoff. Please check the cutoffs.
- Figure 6B: the names on the x-axis needs to be checked.
- The plagiarism rate is very high 34%. Please reduce it.
Author Response
The manuscript by Zhou et al., is very well written in an explained manner. The study emphasized to identify cancer associated fibroblasts (CAFs) related genes in pancreatic cancer (PC) using TCGA and GEO databases via WGCNA approach and identified 26 intersection hub genes. out of which 6 genes GFPT2, MFAP5, CTSK, MMP2, FSTL1, and PRRX1 were established for prognostic assessment and 6 drugs were screened; high-/low-CAF risk groups showed big differences. GSE analysis also showed the enrichment of high CAF risk scores in tumor micro-environment pathways. They concluded that the findings provide new insight and may offer a potential treatment option for chemotherapy and immunotherapy strategies in PC.
The manuscript is very well compiled. But still few points need to be addressed. – We thank the reviewer for the appreciation.
- Figure 2A and 2B: The cutoff seems to be 11 for TCGA-PAAD dataset. But the authors mentioned 8 in line 135. Similarly, in figure 2B, it seems there is no soft threshold power above the cutoff. Please check the cutoffs. We appreciate the reviewer for noticing this. The manuscript has been enhanced through the implemented changes.
- Figure 6B: the names on the x-axis needs to be checked. – We thank the reviewer for identifying the error. The figure has been checked and relabeled.
- The plagiarism rate is very high 34%. Please reduce it. – We thank the reviewer for pointing this out. The manuscript was revised and the percentage of plagiarism was reduced.
Reviewer 2 Report
Comments and Suggestions for Authors
The present article addresses the implications of cancer-associated fibroblast-related gene signatures for facilitating improved prediction of prognosis and responses to therapy in patients with pancreatic cancer. The topic is relevant and seriously evaluated, but major deficiencies identified in both content and form need to be addressed:
-
The abstract is twice as long as the maximum limit allowed in the author guidelines (maximum 200 words). It must be reorganized and reduced by half, highlighting the most important aspects from the background, aim, materials and methods, results, conclusions, and future research directions.
-
Line 19 – “To date, there are no reliable screening tools to detect PDAC at an early stage” – what about the in vitro study of the protein corona?
-
Line 51 – “current” instead of “cur-rent.”
-
Lines 50–76 and 371–421 – the information is organized in the form of an overly long paragraph, which decreases readability and comprehension. Please reorganize it into shorter paragraphs that are more logical and easier to follow.
-
The aim of the paper should be presented in the last paragraph of the introduction and should be addressed and detailed in terms of describing the contribution to the evaluated field and the elements of scientific novelty presented. What was done in the research is already in the manuscript and is also provided in other sections.
-
For the molecular docking study, the selection criteria for ligands and proteins of interest should be better detailed—specifically, why these were chosen over others.
-
Comparisons with similar studies are poorly presented, and the obtained results are insufficiently correlated (or uncorrelated) with findings from the literature.
-
The conclusions should be further detailed, and future research directions should be presented clearly in order to address the limitations of the current study.
Author Response
The present article addresses the implications of cancer-associated fibroblast-related gene signatures for facilitating improved prediction of prognosis and responses to therapy in patients with pancreatic cancer. The topic is relevant and seriously evaluated, but major deficiencies identified in both content and form need to be addressed:
- The abstract is twice as long as the maximum limit allowed in the author guidelines (maximum 200 words). It must be reorganized and reduced by half, highlighting the most important aspects from the background, aim, materials and methods, results, conclusions, and future research directions. – We thank the reviewer fort he suggestion. The word count was reduced, and the text was rendered clear and effective.
- Line 19 – “To date, there are no reliable screening tools to detect PDAC at an early stage” – what about the in vitro study of the protein corona? – We thank the reviewer or bringing this matter to our attention. We have made the necessary edits to address the issue.
- Line 51 – “current” instead of “cur-rent.” – We thank the reviewer for highlighting this inaccuracy. We have rectified it, along with others, to ensure the integrity of the information presented.
- Lines 50–76 and 371–421 – the information is organized in the form of an overly long paragraph, which decreases readability and comprehension. Please reorganize it into shorter paragraphs that are more logical and easier to follow. – We thank the reviewer for his suggestion. The sentence has been shortened and reorganized to enhance its readability and comprehensibility.
- The aim of the paper should be presented in the last paragraph of the introduction and should be addressed and detailed in terms of describing the contribution to the evaluated field and the elements of scientific novelty presented. What was done in the research is already in the manuscript and is also provided in other sections. – We thank the reviewer reviewer for this valuable suggestion. In response, we have incorporated the aim section in the last section of introduction
- For the molecular docking study, the selection criteria for ligands and proteins of interest should be better detailed—specifically, why these were chosen over others. – We thank the reviewer for his insightful remark regarding the molecular docking study. We have incorporated this information into the discussion section.
- Comparisons with similar studies are poorly presented, and the obtained results are insufficiently correlated (or uncorrelated) with findings from the literature. – We thank the reviewer for his concern regarding the suboptimal correlation of our findings with existing literature. A thorough correlation with the extant literature has been completed.
- The conclusions should be further detailed, and future research directions should be presented clearly in order to address the limitations of the current study. – We thank the reviewer for this suggestion. The conclusion has been thoroughly revised to provide a comprehensive overview of the future directions for research.
Round 2
Reviewer 2 Report
Comments and Suggestions for Authors
The authors have significantly improved the manuscript based on the suggestions received.